MCLEAN: Multilevel Clustering Exploration As Network

Alcaide Daniel daniel.alcaide@kuleuven.be daniel.alcaide@esat.kuleuven.be
Aerts Jan
Department of Electrical Engineering (ESAT) STADIUS Center for Dynamical Systems, Signal Processing and Data Analytics, KU Leuven , Leuven , Belgium
imec, KU Leuven , Leuven , Belgium
Kedem Klara
Electronic publication date: 2018 Jan 29
Publication date: 2018
Volume: 4
Electronic Location ID: e145
Received 2017 Dec 4; Accepted 2018 Jan 10
Copyright: ©2018 Alcaide and Aerts
Copyright year: 2018
Copyright holder: Alcaide and Aerts
License: This is an open access article distributed under the terms of the Creative Commons Attribution License, which permits unrestricted use, distribution, reproduction and adaptation in any medium and for any purpose provided that it is properly attributed. For attribution, the original author(s), title, publication source (PeerJ Computer Science) and either DOI or URL of the article must be cited.
License URL: https://creativecommons.org/licenses/by/4.0/

Keywords: Exploratory data analysis, Graph and network visualization, Hierarchical clustering, Visual analytics

Funding: imec strategic funding 2017 IWT SBO Accumulate 150056 KU Leuven CoE PFV/10/016 SymBioSys This research was supported by imec strategic funding 2017, IWT SBO Accumulate 150056, and KU Leuven CoE PFV/10/016 SymBioSys. The funders had no role in study design, data collection and analysis, decision to publish, or preparation of the manuscript.

==============================
Finding useful patterns in datasets has attracted considerable interest in the field of visual analytics. One of the most common tasks is the identification and representation of clusters. However, this is non-trivial in heterogeneous datasets since the data needs to be analyzed from different perspectives. Indeed, highly variable patterns may mask underlying trends in the dataset. Dendrograms are graphical representations resulting from agglomerative hierarchical clustering and provide a framework for viewing the clustering at different levels of detail. However, dendrograms become cluttered when the dataset gets large, and the single cut of the dendrogram to demarcate different clusters can be insufficient in heterogeneous datasets. In this work, we propose a visual analytics methodology called MCLEAN that offers a general approach for guiding the user through the exploration and detection of clusters. Powered by a graph-based transformation of the relational data, it supports a scalable environment for representation of heterogeneous datasets by changing the spatialization. We thereby combine multilevel representations of the clustered dataset with community finding algorithms. Our approach entails displaying the results of the heuristics to users, providing a setting from which to start the exploration and data analysis. To evaluate our proposed approach, we conduct a qualitative user study, where participants are asked to explore a heterogeneous dataset, comparing the results obtained by MCLEAN with the dendrogram. These qualitative results reveal that MCLEAN is an effective way of aiding users in the detection of clusters in heterogeneous datasets. The proposed methodology is implemented in an R package available at https://bitbucket.org/vda-lab/mclean.

Introduction

Determining the number of clusters in a dataset is a frequent problem in data clustering, and is a distinct matter from the algorithm of actually solving the clustering problem. The correct choice of the number of groups is often ambiguous depending on the shape and scale of the points in a dataset and the desired clustering resolution by the user. The optimal choice of clusters depends on the intended use, but in general, it strikes a balance between the maximum compression using a single cluster and the highest resolution of the data by assigning each data point to its own cluster.

Several clustering algorithms have been proposed for partitioning datasets (Jain, Murty & Flynn, 1999). Most of these rely on parameter settings, such as the number of clusters in k-means, the reference value (ε) in DBSCAN or the cutoff distance in a hierarchical clustering. These parameters differ from the algorithm, but either directly or indirectly specify the number of clusters. Setting these parameters demands either detailed pre-existing knowledge of the data or time-consuming trial and error. Moreover, a singular cutoff can hide interesting underlying structures. In the real world, there might not be an single sensible cutoff, and it is common that automatic clustering methodologies ignore particular characteristics of clusters, as some of these might be for example particularly dense or sparse.

As Boudjeloud-Assala et al. (2016) state, “the clustering process is not complete until it is evaluated, validated, and accepted by the user. As such, visual validation and exploration can improve understanding of clustering structure, and can be very effective in revealing trends, highlighting outliers, and showing clusters”. Visualizing clustering results can help to quickly assimilate the information and provide insights that support and complement textual outputs or statistical summaries. Typical questions to be answered regarding clustering results include how well defined the clusters are, how far away they are from each other, what their size is, and if the observations belong strongly to the cluster or only marginally. Therefore the exploration of the different cluster scenarios and the identification of similar record groups (i.e., patterns) in the dataset is a challenge for the user (Vogogias et al., 2016).

Hierarchical clustering is a widely used and effective algorithm to answer these questions, as it provides a framework for viewing the clustering at different levels of detail by imposing a hierarchy on it using a tree (Friedman, Hastie & Tibshirani, 2001). During the cutoff selection process of the tree, the analyst can instantly obtain insights from the graphical representation that suggest the adequacy of the solution but hierarchical clustering does have some drawbacks: (1) the dendrogram representation becomes cluttered when datasets get large; (2) a single cut of the dendrogram is sufficient when the dataset is homogeneous. However, when the dataset is heterogeneous, multiple cuts at different levels might be required. (3) If patterns are present at different levels, choosing a cutoff will hide all but one of these.

Clustering methods often are a fixed process: loading a dataset, setting parameters, running the algorithm, and plotting the results. In other words: clustering is used generally to analyze the data, not to explore it (Boudjeloud-Assala et al., 2016). The integration of visualization and algorithm into the same model is a possible solution to make the clustering process dynamic. The framework to perform interactive visual clustering (IVC) presented by Boudjeloud-Assala et al. (2016) demonstrated a significant advantage in data mining since it allows users to participate in the clustering process by leveraging their visual perception and domain knowledge. As recommended by Keim, Mansmann & Thomas (2010), we believe that if we adapt the visualization environment and combine it with the clustering approach, this combined approach can be used to provide a very natural way for users to explore datasets.

We suggest a novel and generic clustering and exploration approach called MCLEAN (Multilevel Clustering Exploration As Network) for grouping and visualizing multiple granularities of the data that enables: (1) exploration of the dataset using a overview-plus-detail representation, (2) simplification of the dataset using aggregation based on the similarity of data elements, (3) detection of substructures by means of community detection algorithms, and (4) inclusion of the human in the process of selection the number of clusters. Our methodology follows a synergistic approach that combines the strengths of connectivity-based algorithms, community detections techniques and the ability of humans to visually detect patterns, to explore moderately large datasets. It is a visual exploratory and clustering method that permits the user to interact with the algorithm results. The method combines hierarchical clustering algorithms with interactive tools to find optimal clusters and visualize them in a simplified network representation. Network visualizations are an effective means to understand the patterns of interaction between entities, to discover entities with interesting roles, and to identify inherent groups or clusters of entities (Liu, Navathe & Stasko, 2014). The MCLEAN methodology is implemented in an R package available at https://bitbucket.org/vda-lab/mclean.

The remaining part of this paper is organized as follows. In the section ‘Background’ we give an overview of related work in multilevel clustering and graph visualization techniques as an exploratory tool. The section ‘Methods’ describes the proposed visualization technique for clustering exploration in detail, followed by the section ‘Evaluation’, in which we present an evaluation of our approach. Finally, the section ‘Conclusions and Future Work’ presents conclusions and possible directions for future work.

Background

The proposed framework allows the user to employ tacit knowledge in the clustering process in order to detect substructures. This process provides a multilevel environment through overview-plus-detail offering both a general outlook of the data grouping and the precise union of a subset of elements using graphs. To set our work in context, we present a set of examples of visual multilevel clustering and the network transformation of data to identify patterns.

Visual multilevel clustering

There are several methods to perform clustering analysis, but only a few of them support visual analysis. Even fewer provide interactive exploration capabilities of the clusters in different levels of detail. However, the importance of visual interaction for performing clustering analysis is increasingly recognized (Nielsen et al., 2012), as the expert users are capable of steering the analysis to produce more meaningful results. The tacit knowledge often motivates the decisions of the users that algorithms are not able to process or incorporate by themselves. Therefore, including a human in the loop for taking decisions and for guiding the analysis is essential (Vogogias et al., 2016).

Hierarchical clustering has been long used in many different fields including biology, social sciences, and computer vision due to the ease of interpreting the output by the user. The selection of the clusters is based on a single similarity threshold, where the tree is cut at a uniform height. Unfortunately, large and heterogeneous datasets usually require a more flexible approach allowing the user to explore different clustering scenarios. Some methods have been proposed to cut the tree at different levels. Langfelder, Zhang & Horvath (2007) suggested an automatic approach that cuts the branches of the dendrogram in different levels based on their shape. Obulkasim, Meijer & Van de Wiel (2015) proposed a procedure to detect clusters from the dendrogram, called guided piecewise snipping. The method overcomes the drawbacks of the fixed height cut approach by allowing the piecewise rather than the fixed-height cut and incorporating external data to decide upon the optimal cut. In the same line of research, MLCut (Vogogias et al., 2016) is a tool that provides visual support for exploring dendrograms of heterogeneous data sets at different levels of detail.

Partition-based clustering techniques such as k-means and CLARANS (Ng & Han, 2002) attempt to break a data set into k clusters optimizing a given criterion. Boudjeloud-Assala et al. (2016) presented a semi-interactive system for visual data exploration of multidimensional datasets using iterative clustering. Their framework connects the user and the data mining process, which allows the user to play an active role in the clustering tasks. Looney’s approach (Looney, 2002) implements a process of removing small clusters in an iterative way, reassigning them into more dense regions. In doing this, consistency in the clustering results is improved. Similary, Bruneau & Otjacques (2013) proposed an approach to integrate user preferences into the clustering algorithm in an interactively way through 2D projection of the dataset. Rinzivillo et al. (2008) proposed an exploratory methodology for exploring a large number of trajectories using clustering techniques. The grouping of the trajectories is progressively applied by the users refining the parameters of the clustering algorithm.

Graph representation

The dendrogram visual representation is not scalable to larger datasets. A technique presented by Chen, MacEachren & Peuquet (2009) uses a uniform threshold to provide improved visibility by simplifying the dendrogram representation. This is a useful technique for summarising the dendrogram in a selected level of detail and making it fit in smaller displays. However, it does not provide support for multilevel cuts or data exploration at multiple levels.

Given a matrix whose entries represent the similarity between data items, many methods can be used to find a graph representation. In fact, modeling data items as a graph is a common conceptualization used in hierarchical clustering algorithms. In a more general approach, Ploceus (Liu, Navathe & Stasko, 2014) offers an approach for performing multidimensional and multilevel network-based visual analysis on tabular data. Users can flexibly create and transform networks from data tables through a direct manipulation interface. Ploceus integrates dynamic network manipulation with visual exploration for a seamless analytic experience.

The WhatsOnWeb system (Di Giacomo et al., 2007) takes advantage of graph-based visualization created by the results of a Web search engine. In their system, a search query produces a graph that represents sets of Web pages as nodes, which are connected if documents are sufficiently semantically related. The strength of the relationship is encoded with an edge weight and a topological clustering algorithm is recursively applied to the graph, forming a graph hierarchy and showing different levels of information.

Systems presented for clustering and exploration in Duman, Healing & Ghanea-Hercock (2009), Desjardins, MacGlashan & Ferraioli (2007), Beale (2007) and Lee et al. (2012) transform the data into a spring-embedded graph layout, encoding the distance between the elements as forces in the force-directed layout. The objective in these systems is the projection of the distances in a reduced dimension allowing clustering assignment using partitioning-based methods. Links are usually omitted in the representation facilitating the readability of the spatialization of the nodes. They present an alternative to standard dimension reduction methods such as projection pursuit or multi-dimensional scaling.

The network exploration of MCLEAN can be considered close to the solutions proposed for the navigation of the clustering results for large-scale graph visualization systems, such as Eades & Feng (1996) and Eades & Huang (2000). They allow the user to navigate a graph by iteratively expanding or collapsing the aggregated nodes (meta-nodes). However, users often lose context when navigating clustered graphs with deeper hierarchies (Abello, Van Ham & Krishnan, 2006).

Methods

The MCLEAN method takes a similarity matrix of all data records as input, and produces a simplified graph representation showing a higher abstraction of the clustering process. MCLEAN combines two visual representations. First, an overview plot (barcode-tree), related to a dendrogram and topological barcode plot, shows how the general cluster structure changes for different values of a parameter ε, indicating how close points need to be in the multi-dimensional space to be considered belonging to the same cluster. Second, a node-link plot represents the clustering results at a given ε. For this ε, clustering information in the node-link diagram is dual-layered. First, graph connected components correspond to data clusters at this threshold ε. Second, different colours within a connected component indicate that this subnetwork would be split when using a more stringent ε; in other words, it indicates substructures in this cluster.

A connected component is a subgraph in which all the vertices are directly or indirectly connected. We use connected components to define the clusters in the dataset. In addition, MCLEAN employs community detection algorithm to find subclusters inside connected components. As a result, user knowledge (tacit or other) can inform on whether a cluster is distinct or is a part of a larger cluster. This ambiguity is common in heterogeneous data sets.

As most of the clustering techniques, the agglomerative algorithm that we use depends on one single parameter. This parameter is a threshold (ϵ) that defines the distance of union between two data elements. We find similarities between the MCLEAN approach and topological data analysis (TDA). Analyzing the multidimensional spaces from a topological structure perspective, interpreting the persistent homology by calculating the number of connected components (b0 from betti numbers) and using the persistence concept to define the optimal threshold of network representation prove that although the aims are distinct they share a same philosophy of analysis (Topaz, Ziegelmeier & Halverson, 2015).

The MCLEAN method consists of four parts as illustrated in Fig. 1: (1) transformation of the distance matrix into a node-link representation based on the threshold defined; (2) simplification of the network creating aggregated nodes; (3) detection of substructures employing community detection algorithms; and (4) exploration of the resulting networks for different threshold values.

Figure 1 Workflow diagram of MCLEAN algorithm, consisting of four steps: (1) graph transformation, (2) node aggregation, (3) community detection and, (4) barcode-tree creation.

The methodology in this section is illustrated using a dataset taken from the UCI repository website (see Fig. 2). This dataset contains 600 examples of control charts synthetically generated as described by Alcock & Manolopoulos (1999). We used Dynamic Time Warping (DTW) for measuring similarity between the temporal sequences. Figure 2 illustrates both representations of the raw data (Figs. 2A and 2B) and classical visualizations of the distance matrix such as the dendrogram (Fig. 2C) and a scatterplot of the two first dimensions of multidimensional scaling (Fig. 2D).

Figure 2 Representation of a synthetic dataset that contains 600 examples of control charts synthetically generated by the process in Alcock & Manolopoulos (1999).

(A) Time series are treated as a unique group. (B) Underlying (hidden) time series are split by their label. (C) Dendrogram using single linkage. (D) Two first dimensions of classical multidimensional scaling. Color represents the label in the dataset.

Graph transformation

Multidimensional Scaling (MDS) projects the data elements in reduced dimension ordination space. Two or three dimensions are often used, which is based on ease of visualization rather than on the dimensionality of substructures in the data. Unfortunately, in some cases, these projections blur patterns due to the heterogeneity of the distances and the limitations of the space visualized. Therefore, a change to the spatialization (such as network visualization) can help to overcome the limitations of complex datasets. An example of these weaknesses can be seen in the MDS applied to the Synthetic dataset in Fig. 2D.

Although the distance matrix does not contain explicit network semantics, MCLEAN uses this approach to transform the encoding of distances by the use of links in the network. Moreover, the algorithm employed in the final drawing of the network (i.e., force-directed graph) is optimized to avoid overlapping between the nodes.

The graph transformation step of MCLEAN is similar to the DBSCAN method (Ester et al., 1996), in that it relies on a parameter ϵ which defines the radius that designates points to be lying in each other’s neighbourhood. In DBSCAN, a second parameter numPts is used to define the minimal number of points that can constitute a cluster. In MCLEAN, however, all datapoints are considered network nodes, and datapoints that are within a distance ϵ from each other are linked. The result of this step produces a graph where there exists a path between two nodes if and only if they belong to the same connected component. At this stage of the methodology clusters are represented as connected components in topological space. Figure 3 shows the graph transformation process for four snapshots of different parameters ϵ applied to the same dataset. As ϵ increases, the number of links grows between the nodes.

Figure 3 Node-link network transformation using force-directed layout from the distance matrix using a distance threshold of 150 in part (A), 190 in part (B), 220 in part (C) and, 290 in part (D).

All data elements are represented as a node in the network. Edges are defined based on the threshold.

Node aggregation

In case of large datasets, the node-link representation can become visually overwhelming for the user without a proper level of aggregation. The challenge is to extract understandable information buried in the structure of multiple nodes and links. In addition, a layout of the entire graph is costly to compute. MCLEAN simplifies and highlights the structure of the raw network. This process of simplification is founded on the use of aggregating nodes (meta-nodes) that represent a subgraph at a higher level of abstraction.

Node aggregation is based on degree centrality, where the degree of a node is defined as the number of connections that the node has within a network. This value is computed for all nodes, and the highest one is the first candidate to be the center of an aggregated node (meta-node). All nodes connected directly with the candidate are converted into an aggregated node. All connections with other data elements are inherited in the meta-node keeping the structure of the connected component. The result of node aggregation for the graphs created in Fig. 3 is shown in Fig. 4.

Figure 4 Network representation of the clustered dataset using a parameter ε of 150 in part (A), 190 in part (B), 220 in part (C) and, 290 in part (D).

This representation preserves the same structure shown in Fig. 3.

Our simplification graph approach was designed to preserve the structure of the input graph. According to Archambault, Munzner & Auber (2008), a topologically preserving graph must respect the following two properties:

1. Edge Conservation: an edge exists between two meta-nodes m1 and m2 if and only if there exists an edge between two leaves in the input graph l1 and l2 such that l1 is a descendant of m1 and l2 is a descendant of m2.

2. Connectivity Conservation: any subgraph contained inside a meta-node must be connected.

By respecting these two properties, we ensure that the resulting graph preserves the topological features of the initial graph: edge conservation guarantees that any edge in the simplified graph is present in the initial graph, while connectivity conservation ensures that any path can continue through any meta-node (Archambault, Munzner & Auber, 2009).

In MCLEAN, meta-nodes are created through the densest nodes (highest degree) in a connected component. The node with the highest degree is the best candidate to be the center of the meta-node. Figure 5A shows an illustration of a connected component where node eight is the best candidate. All nodes connected directly to the best candidate become part of the meta-node as shown in Fig. 5B. A meta-node inherits the edges with the external nodes or meta-nodes that do not belong to it. The aggregation is an iterative process until all nodes become part of a meta-node. Aggregated nodes are excluded from the process preventing them to be included into another meta-node. For example node ten is part of the meta-node of node eight. Therefore, it cannot be included in or be a candidate for a new meta-node although it has the same degree as four and fourteen in Fig. 5B. The number of connected components or clusters does not change after the simplification process. Figure 5B show the result of the node aggregation process.

Figure 5 Illustration of node aggregation process for a set of fifteen elements.

(A) Network without aggregation. Node 8 in the network is the best candidate to build the first meta-node. All nodes directly connected to this node become part of the meta-node. (B) Network after the creation of the first meta-node. The aggregation continues until all nodes are aggregated. The best candidates are node 4 and node 14 at this step. (C) Network at last stage of the aggregation process. Node 1 and 6 become individual meta-nodes. (D) Resulting aggregated network.

Community detection

The simplified network representation (Fig. 4) preserves structural data in a compressed way, which together with community detection allows revealing substructures inside connected components. A community refers to a group of nodes that are internally highly connected. Community detection in networks is not a trivial problem, and many algorithms have been proposed. MCLEAN relies on the Infomap algorithm (Information-theoretic method) (Rosvall & Bergstrom, 2008), which provides multilevel solutions for analyzing undirected, directed, unweighted, and weighted networks. In MCLEAN, the number of data elements in each simplified node is used as vertex weight in the Infomap algorithm to reduce the effect of aggregation. Different communities in a single connected component are shown in different colors. Figure 6 shows the networks created after graph transformation (Fig. 3) and node aggregation (Fig. 4) applying the results of community detection. Prevalence of communities increases with network size, as shown in Fig. 6 where part A does not reveal any substructure but part D shows three in a single connected component.

Figure 6 Network representation of the clustered dataset using the distance threshold of 150 in part (A), 190 in part (B), 220 in part (C) and 290 in part (D).

Communities are detected through Infomap. The coloring of nodes illustrates the communities detected by the algorithm.

Barcode-tree

As indicated above, the generated connected components depend on the value of parameter ϵ, as can be seen in the four subplots in Fig. 3. In general, the network consists of isolated vertices for small values of the threshold. At the largest value, the entire dataset is a single connected component. The selection of a representative threshold without prior knowledge of the underlying space is however difficult for any dataset. In addition, heterogeneous datasets may need multiple levels of partitions and therefore will require the exploration of multiple thresholds.

In order to provide guidance in the parameter choice and a contextual overview of the relation between ϵ and clustering results, MCLEAN generates these graphs across a range of ϵ values. These are subsequently combined in the tree representation called barcode-tree, which is inspired by both a clustering dendrogram and barcode representation (Topaz, Ziegelmeier & Halverson, 2015) as used in topological data analysis.

The barcode-tree (Fig. 7) is a visual representation of cluster arrangement. The horizontal axis corresponds to threshold ϵ and refers to the distance measure of union between the data elements that define the network (see ‘Graph transformation’ section). The individual components are arranged along the vertical axis of the plot. At any given threshold, the number of connected components is the number of lines that intersect the vertical line through the threshold. Meta-nodes are formed in the join points that are aggregations of individual data elements or existing meta-nodes at a smaller threshold (see ‘Node aggregation’ section). This tree overcomes the limitations of binary structure of a dendrogram, allowing for a more clear representation of branches. Moreover, the barcode-tree implements a leaf ordering method motivated by the MOLO algorithm presented by Sakai et al. (2014). The branches are evaluated backwards recursively (from the single cluster until the singleton) to be the center of the subtree at each threshold avoiding the crossing of the branches.

Figure 7 Barcode-tree for a sequence of thresholds from 0 to 300 by steps of 5 using gradient color to represent the number of communities for each connected component.

Meta-nodes for a small threshold are aggregated into new ones created by the larger threshold: if ϵ1 ≤ ϵ2 ≤ ϵ3 ≤ ⋯ ≤ ϵN−1 ≤ ϵN then M1⊆M2⊆M3⊆⋯⊆MN−1⊆MN with Mi being the meta-nodes in network i. If the user is interested in understanding the structure of the input data, then topological hierarchies are useful tools to explain the origin of all edges viewed in a cut. Both the objective for the barcode-tree view in MCLEAN and the barcode in TDA is to find the persistent topological structures across a range of thresholds. Those structures which persist over an extensive range are considered signals of the underlying topology. As the threshold changes, the topological structures of network change accordingly.

In Fig. 7, we see the representation of the connected component for the range of ϵ from 0 to 300. For ϵ = 100, we see 600 connected components because there are no connections amongst the individual elements in the dataset. For ϵ = 220, we see a big connected component and a significant subset of individual elements, reflecting the fact that some vertices have joined into a larger connected component. For ϵ = 290, we see a single connected component that indicates the joining of all data elements.

The resolution of the plot depends on the number of evaluations, showing a general overview with only a few thresholds (Fig. 8E) or allowing detailed understanding of connected component composition with a more dense covering of thresholds (Fig. 8A).

Figure 8 Set of five curves of connected components vs. threshold distance according to different granularities.

(A) barcode-tree for the range of ε (0 to 300) by steps of 1, (B) by steps of 2, (C) by steps of 5, (D) by steps of 10 and (E) by steps of 15 and (F) by steps of 20.

Although the exploration of the connected components path around a threshold of interest can give an intuition of the resulting network, the analysis of the connected components using the network representation (Fig. 6) is a necessary step to identify hidden substructures using only the tree representation. For example, at threshold 290 in Fig. 7, we identify a single connected component. However, we identify more details in the structure of the network at Fig. 6D.

Evaluation

To understand the implications of the proposed methodology and the interaction between the visuals, we performed a qualitative evaluation regarding learnability and usability of MCLEAN. We recruited six participants, including four doctoral and two post-doctoral researchers in the area of data science with knowledge of clustering techniques (e.g., hierarchical or k-means clustering) and dimensionality reduction techniques (e.g., multidimensional scaling and PCA). None had seen or used MCLEAN before the evaluation test. The goal of the evaluation was to identify qualitative insights about how well MCLEAN supports the identification of patterns according to the simplification of the dataset.

Tasks and procedures

We gave a brief introduction to MCLEAN, explaining the fundamentals of the methodology and demonstrating the main functionalities of the interface developed to interact with the network and barcode-tree including the bidirectional selection of elements between the two visuals (see Fig. 2). A training exercise was performed to familiarize the participants with the MCLEAN workflow using the Fisher Iris dataset (Fisher & Marshall, 1936). We asked the participants to explore the general patterns in the barcode-tree and specific topological structures utilizing the network representation and community finding results. We repeated the exercise for the actual evaluation over the control charts dataset, explaining only that it concerned time-series data. We asked the participants to think aloud, observed their interaction with the interface, recorded their patterns selection as hand-written notes, and sought their impressions and comments on the methodology after they completed the tasks. To conclude we asked them to complete a questionnaire to evaluate the efficacy and their satisfaction of MCLEAN compared to dendrogram. We also sought to know how difficult the methodology was to learn and use, if there were any problematic design issues, and how we might be able to address the difficulties experienced by the participants.

Results and analysis

The evaluation exercise was split into three parts: detection of patterns using the barcode-tree, selection of thresholds comparing the dendrogram and barcode-tree, and detection of patterns combining the network representation and barcode-tree. After the exercise, the questionnaire was provided to obtain user satisfaction and additional feedback.

Detection of patterns using the barcode-tree

In a first evaluation, we sought to identify to what extent participants are able to identify the different underlying patterns as shown in Fig. 2B. Five participants (participants A–E) identified four different patterns in the temporal dataset using the barcode-tree exclusively for the range of ϵ 0 to 300 by steps of 5 (Fig. 7). Participant F identified two patterns using the same representation, grouping pattern A1, A2 and A3 as a single pattern and pattern A4 independently from the rest (see Fig. 9). Identical results were found in the dendrogram exploration. In addition, pattern A4 was classified as a group of outliers by participants D and E.

Figure 9 Detection of patterns between the barcode-tree and dendrogram.

Part (A) highlights the four patterns detect by five out of six participants in the barcode-tree. Part (B) shows the four patterns detected in the dendrogram. Pattern B1* was identified as an additional pattern by two out of five participants.

When identifying four patterns, users A-E aggregated Type 3 and Type 5 (see Fig. 2B) signals into a single pattern (pattern A2; see Fig. 9A), and Type 4 and Type 6 in pattern A3. In both cases, the pairs behave similarly, but in opposing directions: a continually increasing or decreasing trend, or a shift in the middle of the time series. In each pair, the global distance between the different types is small compared to the rest of patterns due to the sequence alignment using DTW.

When using the dendrograms, (Fig. 9B), participants A–E identified between three and five patterns. Two or three groups were detected in the middle, and the heterogeneous data elements (pattern B1) were recognized as an additional pattern and misunderstood as two independent clusters by users B and E due to the location of the branches. Participant C identified a single cluster containing signals of Type 1, Type 3 and Type 5, and another containing Type 4 and Type 6. This result shows a slight loss of perception in the dendrogram compared to barcode-tree and a possible potential misinterpretation of the dendrogram due to to the position of the branches.

Changes in tree resolution did not present a change in the interpretation of participants when resolution was increased, i.e., steps of 1 and 2 (Figs. 8A and 8B) but it did when the resolution decreased. Three participants (B, C and E) detected six patterns when we evaluated the number of connected component in steps of 20 as shown in Fig. 8F. This fact reveals that different resolutions lead to different possible interpretations of the data.

Selection of cutoffs in dendrogram and barcode-tree

Using dendrogram exploration, only participant F experienced difficulties in cutoff selection, whereas participants A-E selected a single cutoff between thresholds 180 and 195, describing two or three notable clusters and ungrouped data-elements. Using the barcode-tree, participants A-E selected a similar threshold. Participant D investigated an additional threshold at 220. Participant F picked the threshold 285 with the intention of exploring the network representation. The number of cutoffs was not limited in any of the representations allowing the user to explore different partitioning perspectives. Overall, users were more confident in choosing thresholds using the barcode-tree than when using the dendrogram. A persistent segment starts at the ϵ threshold of 185 until the join of three clusters at threshold 202. Discussion with the participants indicated that this persistence in the barcode-tree makes for better readibility and therefore higher confidence in threshold selection. In contrast, the binary union of the branches and non-optimisation of the leave ordering in the dendrogram can lead to misinterpretation of the cutoff selection leaving some elements outside of a potential cluster.

Detection of patterns combining the network representation and barcode-tree

In this part, we aimed to evaluate the detection of the structures through community detection and interaction between the visualizations. We identified three relevant thresholds for the network representation at the start of the three most persistent topological structures (see Fig. 9A), more specifically at threshold 187, 202, and 283, and invited the participants to describe the patterns seen using both the network and the tree representations (Fig. 10). We encouraged them to use the interaction between these visual encoding to clarify their relationship. Three participants (C, E and, F) identified six patterns at threshold 187, while the others recognized four. The number of patterns recognized was three for all participants at thresholds 202 and 283. Both the network representation and the color-encoding to represent the communities detected by Infomap were clear for all participants. The difference in number of perceived patterns shows a critical sense of the community detection results, demonstrating the added value of the human in the pattern selection.

Figure 10 Network representations of the synthetic time-series dataset (Fig. 2) at the beginning of the three most persistent structures detected in the barcode-tree.

(A) Network at threshold 187. The grey node corresponds to Type 1 signal. The two connected components correspond to signals of Type 3/5 (blue and red) and Type 4/6 (orange and green), respectively. In both cases, ascending—respectively descending—signals are combined in a single connected component but still distinguish between gradual or stepwise change based on colour. (B) Network at threshold 202. The most significant connected component in the network integrates all signals in the dataset, excluding Type 2 represented as individual grey elements. Two communities represented as different colors distinguish the ascending (blue) and descending (orange) patterns of the network. (C) Network at threshold 283. The single connected component network still allows the detection of the ascending and descending patterns and the high variability of signal Type 2 (green) due to the community detection algorithm.

User satisfaction and comments

All the participants indicated that they liked the MCLEAN methodology, especially the obtained interpretation due to the change of the layout in the network creation. Although some participants considered the selection of thresholds and interpretation of community detections nontrivial, they still agreed that the methodology was consistent and the learning curve was not too high. Participants strongly favored the use of MCLEAN over dendrogram in terms exploration and clustering technique due to the better readability of the tree and the power of combining the two visualizations interactively. This indicates the benefits of this methodology as an interactive visual clustering facilitating the integration and evaluation of the results by the user.

Conclusion and future work

In this paper, we described a method for interactive multi-resolution exploration of clustering results in complex datasets. Evaluation experiments indicated that combining visualizations and analytical techniques can increase the understanding of the information for the user by providing more transparency and confidence to the process. Although the number of clusters and their quality is strongly related to user behavior, we believe that this is actually a strength of the system (one that was specifically aimed for), and that these approaches used in conjunction are crucial to allowing a user-centric approach to information discovery, to exploit heterogeneous data sources better.

Although the presented network and barcode-tree representations help the user in gaining insight in their data, there are some clear points for future work. For example, the current approach relies on single linkage clustering whereas average and/or complete linkage clustering might be more useful for particular datasets (especially where the distance matrix does not exhibit gaps). In addition, the current visual encoding of the barcode-tree shows visual artifacts (parallel lines merging with a cluster) depending on the granularity level used. Finally, it will be useful to investigate further methods for directly comparing how data elements are integrated across thresholds.

In conclusion, incorporating the domain user in the clustering process itself allows for retaining the richness of multilevel patterns in cluster results. MCLEAN facilitates integrating tacit or other user knowledge in clustering result interpretation and exploration, while simplifying the representation of groups especially in the presence of noise or outliers. We argue that the MCLEAN approach provides new opportunities beyond existing techniques for cluster visualization and exploration.

The authors wish to thank Houda Lamqaddam for guidance in the design of the evaluation, and the evaluation participants for valuable feedback.

Additional Information and Declarations

Competing Interests

Author Contributions

Data Availability

The authors declare there are no competing interests.

Daniel Alcaide conceived and designed the experiments, performed the experiments, analyzed the data, wrote the paper, prepared figures and/or tables, performed the computation work, reviewed drafts of the paper.

Jan Aerts conceived and designed the experiments, analyzed the data, wrote the paper, reviewed drafts of the paper.

The following information was supplied regarding data availability:

Bitbucket: https://bitbucket.org/vda-lab/mclean.

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
