# Peer review of "MCLEAN: Multilevel Clustering Exploration As Network"

_PeerJ Computer Science, doi:10.7717/peerj-cs.145_

## Round 0.1 · accepted · Accept

Two reviewers have provided very favorable comments.

The paper presents a novel method for interactively exploring complex datasets in order to discover patterns and clusters. The novelty lies in the combination of computational analysis and visual interpretation. In view of the plethora of ever more detailed and complex data becoming available in many fields, a tool to aid in analysis is very relevant.
The article is up-to-date and well structured with references form the last years and clearly presents the advantages of the study performed.

A suggestion for possible improvement: To the inexperienced user it is not obvious how the network representation of the clustered dataset (figure 4) results from the node-link network transformation (figure 3). Maybe it would help to orient the figures such that the correspondences between aggregated nodes and the original clusters can be more easily identified.

·

Basic reporting

The paper is very well written and thus straightforward to read: The English is clear and unambiguous, literature references and background are provided and the given figures are helpful to support the information presented in the text.
Just one idea for a possible improvement: To the inexperienced user it not obvious how the network representation of the clustered dataset (figure 4) results from the node-link network transformation (figure 3). Maybe it would help to orient the figures such that the correspondences between aggregated nodes and the original clusters can be more easily identified.

Experimental design

The paper presents a novel method for interactively exploring complex datasets in order to discover patterns and clusters. The novelty lies in the combination of computational analysis and visual interpretation. In view of the plethora of ever more detailed and complex data becoming available in many fields, a tool to aid in analysis is very relevant.
The paper adequately explains the underlying computational methods and demonstrates the capabilities of the new tool using a suitable data set.

Validity of the findings

The authors validate their tool by studying the usage by members of their target audience. Thus, they can establish that the new tool has the desired advantages over state of the art methods.

·

Basic reporting

The article is up-to-date and well structured with references form the last years and present clearly the advantages of the study performed.

Experimental design

In an efficient manner the methodology aims to highlight the multilevel representations of the clustered dataset with community finding algorithms.

Validity of the findings

The article exposes clearly the results of the study performed.

Additional comments

The article is up-to-date and well structured with references form the last years and present clearly the advantages of the study performed.
In an efficient manner the methodology aims to highlight the multilevel representations of the clustered dataset with community finding algorithms.
The article exposes clearly the results of the study performed.